# Effective Hospital Care Delivery Model for Older People in Nigeria with Multimorbidity: Recommendations for Practice

**DOI:** 10.3390/healthcare10071268

**Published:** 2022-07-07

**Authors:** Abdulsalam Ahmed, Hafiz T. A. Khan, Muili Lawal

**Affiliations:** College of Nursing, Midwifery, and Healthcare, University of West London, Paragon House, Boston Manor Road, Brentford TW8 9GB, UK; hafiz.khan@uwl.ac.uk (H.T.A.K.); muili.lawal@uwl.ac.uk (M.L.)

**Keywords:** multimorbidity, healthcare, quality, patients, Donabedian model, patients experience and satisfaction

## Abstract

The importance of developing an effective action-based model of care for multimorbid patients has become common knowledge, but it remains unclear why researchers in Nigeria have not paid attention to the issue. Hence, this study assessed the quality of health services using the Donabedian model and aimed to recommend an effective hospital care delivery model for older people in Nigeria with multimorbidity. A cross-sectional study using face-to-face data was conducted between October 2021 and February 2022. The reported data were collated, checked, coded, and entered into JISC online survey software and then exported to IBM Statistical Package for Social Science (SPSS) version 27 for analysis, sourced from the University of West London, London, United Kingdom. The data were collected from the outpatient department of four high-volume public secondary hospitals in Niger State (the largest hospital in the three senatorial zones and that of the state capital). Systematic random sampling was used to select 734 patients with two or more chronic diseases (multimorbidity) aged 60 years and above who presented for routine ambulatory outpatient and consented to participate in the study. A Service Availability and Readiness Assessment (SARA) tool was used to assess the structure, and the process quality was assessed by the patients’ experiences as they navigated the care pathway, whereas the outcome was measured using the patients’ overall satisfaction. Using Spearman’s correlation, no statistically significant association was observed between satisfaction level with the healthcare that was received and the five domains of health facility readiness (Total score Basic Amenities, Total score Basic Equipment, Total score infection control, Total score diagnostic capacity, Total score essential drugs), and the general facility readiness. Finally, the process component superseded the structure as the determinant of the quality of healthcare among multimorbid patients in Niger State. The emphasis of the process should be on improving access to quality of care, improving patient–physician relationships and timing, reducing the financial burden of medical care, and building confidence and trust in medical care. Therefore, these factors should be incorporated into designing the healthcare model for multimorbid patients in Nigeria.

## 1. Introduction

Multimorbidity is a growing global health issue that is likely to become challenging in developing countries such as Nigeria as they experience economic, demographic, and epidemiological transitions. One major barrier to the quality of healthcare for most of the population in these developing countries is the lack of access to even basic health services [1], not to mention quality care. The quality of care has been defined as the degree to which health services for individuals and populations increase the likelihood of desired health outcomes and are consistent with current professional knowledge [2]. What remains of concern is how to successfully measure the quality of care in the general health care setting, as well as for patients with two or more chronic diseases who usually have complex health care needs. Health care quality is defined not by a patient’s ultimate health destination, but rather by the distance that is traversed in conveying the patient to that destination [3]. In Nigeria, despite efforts by the government through the Ministry of Health to improve the quality of care through different approaches such as universal health coverage, and various health strengthening strategies such as the formation, analysis, and revision of policies, guidelines, protocols, and strategies for sustainable healthcare financing in Nigeria, health service provision is constrained by a number of factors in terms of poor infrastructure, the unavailability of drugs and/or medical equipment, and limited human resources for healthcare [4,5].

Nigeria operates a pluralistic healthcare delivery system (orthodox and traditional healthcare delivery systems). Orthodox health care services are provided by the private and public sectors. However, the provision of health care in Nigeria remains the function of the three tiers of government: the federal, state, and local government. Multimorbidity challenges existing healthcare organizations, research, family relationships, and social security [6], chiefly because the healthcare organization remains a single- disease/condition focused organization.

Although it has been reported that multimorbidity should promote a shift in the way that health policies are developed and guide the healthcare system in tackling this challenge, one big, persistent issue is the care for multimorbidity, because of the associated complex healthcare needs [7]. Similarly, studies have argued that limited research on multimorbidity, especially in developing countries such as Nigeria, curtails the development and implementation of sustainable healthcare models [8]. Therefore, a better understanding of the quality of care that is available to multimorbid patients is imperative in the process of selecting new interventions and building strategies for the quality improvement of them. Although choosing a measure of multimorbidity mainly depends on the suitability of the measure for the data that is obtainable and the predilection of the researcher, studies have reported that the most common approach to measure multimorbidity is the use of simple disease counts [9]. Further, they concluded that simple measures, such as counts of chronic diseases, are almost as effective at predicting health care utilization and quality of life as more sophisticated measurements. On account of the aforementioned reasons, this study uses a simple disease count.

It is suggested that patient satisfaction is affected by the attitude of health workers toward patients, their ability to offer immediate attention, waiting time, ability to send information, and the tolerance of physicians to plainly explain to the patient what is wrong before giving a detailed message concerning their drugs and the environment [10]. Studies have also shown that there are many factors affecting the satisfaction and dissatisfaction of patients in hospital facilities. These factors include access, health personnel, financing, the cleanliness of the health facility, patient-physician time, and patient waiting time [11].

There is increasing awareness of the perception of quality of care as an important driver of care [12,13], while understanding that patient experience is a key step in moving toward patient-centered care. That quality-of-care assessment can provide a critical starting point to develop an effective action-based model of care for multimorbid patients has become common knowledge, but it remains unknown why researchers in Nigeria have not paid attention to this issue. The main objective of the study was therefore to recommend an effective hospital care delivery model for older people in Nigeria with multimorbidities by using the Donabedian model of care to examine the effectiveness of the current care pathway setting for multimorbid patients in Niger State, Nigeria. See Figure 1.

## 2. Materials and Methods

A cross-sectional study was conducted between October 2021 and February 2022 using face-to-face data that were collected and entered into JISC online survey software. With the utilization of the statistical literature, a total sample size (*n*) of 800 was determined with an adequate statistical formula. A purposeful sampling method was used to select 4 high-volume general hospitals, one in each of the 3 senatorial districts and one in the state capital, all with a good representative of multimorbid patients. Systematic random sampling was used to select 734 patients with 2 or more chronic diseases (multimorbidity) aged 60 years and above who presented for routine ambulatory outpatient and consented to participate in the study. The participants were uninsured and recruited from the outpatient department of 4 secondary hospitals: General Hospital Minna, General Hospital Bida, General Hospital Suleja, and General Hospital Kontagora. General Hospital Minna is in the state capital and is the largest state-owned health facility in Niger State. The general hospitals in Bida, Suleja, and Kontagora are the largest hospitals in the senatorial zones A, B, and C, respectively. The study was limited to 4 secondary hospitals in the state that served as referral centers for the primary health institutions, private facilities, and other secondary hospitals of their respective zones across the 25 LGAs in the state. These 4 hospitals combined around 85 to 90% of the patients in the state.

Data collection was carried out with the use of a structured, pre-coded questionnaire that was administered by the principal investigator and trained assistants as the patients left the hospital after consultation on scheduled clinic days. Patients’ consent was sought and they were given the opportunity to consider participating with at least a 24-h gap between provision of the study information and being involved in the interview. While some patients consented immediately to the interview, others did not consent immediately and were given 24 h to make their decision. After obtaining written informed consent, the participants were interviewed face-to-face by the researcher and trained research assistant using a pre-validated structured questionnaire. The participants were not compensated but they were assured of the confidentiality of the information that they provided. The survey interview was conducted in English or Hausa language, the most popular language in Niger State, Nigeria (whichever the respondent felt comfortable with).

Ethics statement: Ethical approval was obtained from the College of Nursing, Midwifery, and Healthcare, Research Ethics Panel, University of West London (Ethical Approval No. 1055), and authorization to collect data was obtained from the Research, Ethics, and Publication Committee (REPC) of the Hospitals Management Board, Minna, Niger State, Nigeria. The participants freely signed their informed consent about 24 h prior to participating in the study, and the individual’s right to withdraw partially or completely was observed.

### 2.1. Measurement of Variables

To provide comprehensive information on the quality of care that the multimorbid patient received in the hospitals, the Donabedian model of care was used. This model was a good fit and was adopted for this research because it explored all three elements of quality of care (structure, process, and outcome). The Service Availability and Readiness Assessment (SARA) tool was used to assess the structure, and the process quality was assessed by the patients’ experiences as they navigated the care pathway, whereas the outcome was measured using the overall patient satisfaction. The SARA facility assessment tool focused on an inventory of availability and readiness of basic health facility structures: basic amenities (7 tracer items); basic equipment (6 tracer items); diagnostic capacity (8 tracer items); essential drugs (20 tracer items); and standard precaution for infection prevention (9 tracer items). For the process and outcome quality measurement, the patient satisfaction questionnaire (PSQ)-18 was adopted [14]. This is the revised short-form version of PSQ-III and PSQ but it retains many of the characteristics of its full-length counterpart, including general satisfaction, technical quality, interpersonal communication, financial aspects, time spent with the doctor, accessibility, and convenience.

### 2.2. Statistical Analysis

The reported data were collated, checked, coded, and entered into JISC online survey software and then exported to SPSS version 27. The data were then cleaned and analyzed using descriptive and inferential statistics. The descriptive statistics were used to summarize the overall characteristics of the participants including gender, age, marital status, family structure, education level, ethnicity, occupation, and level of income. A descriptive and comparative statistical data analysis was processed to answer the research objective.

The general service readiness was assessed by using the five domains of tracer indicators: (1) basic amenities; (2) basic equipment; (3) standard precaution for infection prevention; (4) diagnostic capacity; (5) essential drugs. The average readiness score for each domain tracer was calculated by the ratio of the available tracer item over the total required items. The average service readiness index for each health facility was determined by adding the mean score of the five domains and dividing by 5 (total number of the domains). To assess general service readiness, we first calculated scores for each of the five domains (amenities, basic equipment, infection prevention, diagnostic capacity, and essential medicines) based on the mean availability of tracer items as a percentage within the domain. Then mean of all five domains was calculated and expressed as a general service readiness index. Each domain carries equal weight and the average general service readiness score represents the overall readiness status of the hospitals to provide services.

The assumptions were met for Spearman’s correlation and the relationship between satisfaction level with the healthcare that was received, and the five domains of health facility readiness and the general facility readiness was tested. Assumptions were also met for linear regression, and the overall satisfaction level with medical care was predicted using linear regression (patience experience as an independent variable). In the first model, a simple regression was performed, and most of the multimorbid variables of patient experience were predictors of overall satisfaction with healthcare. A principal component analysis (PCA) was used to reduce the dimensions of the multimorbid patient experience into main components, because assumptions were met for PCA. The analysis showed that the data met the assumptions of sample adequacy (KMO = 0.87), the absence of multicollinearity (r < 0.6), and the significance of Bartlett’s test of sphericity. Factor extraction established four components with eigenvalues that were greater than one. The analysis of how each question loaded onto different components revealed varied themes in the questions. Although many techniques have been developed for dimension reduction, the principal component analysis (PCA) is used in this study, despite being one of the oldest, not only because it is the most widely used, but also because its idea is simple and reduces the dimensionality of a dataset, while preserving as much ‘variability’ (i.e., statistical information) as possible [15].

## 3. Results

### 3.1. Characteristics of the Participants

A total of 800 patients aged 60 years and above were approached for inclusion in the study and 91.8% (734 out of 800) agreed to participate. A total of 66 patients refused to participate for personal reasons. All four secondary health facilities attained or surpassed the minimum required sample size. About 60% of the respondents were female and the mean age of the sample was 67.3 years (male 66.3 years and female 68.1 years)—see Table 1. The most frequent marital status was married in 65.8% of the sample. The major family structure was extended family in 60% of respondents. A considerable proportion of the respondents did not have any form of education (62.9%) and owned a business as their occupation (38.1%). The majority of the respondents were from the major ethnic groups of the state (Nupe 27.8%, Gwarri 26.3%, and Hausa 23.7%). Less than NGN 15k was reported in nearly two-thirds of the cases, which was less than USD 36 at the official rate of USD 1 to NGN 414.52 (27 June 2022).

### 3.2. Readiness Status in Five Domains

General service readiness is described by the following five domains of tracer indicators: (1) basic amenities; (2) basic equipment; (3) standard precautions for infection prevention; (4) diagnostic capacity; (5) essential medicines. The average readiness score for basic amenities was the same for general hospitals in Kontagora, Minna, and Suleja with an 85.7% score, and the lowest in General Hospital Bida with a 28.6% score (see Table 2). The average basic amenities readiness score was 100% across all the hospitals. The average score for standard precautions for infection prevention readiness measures in the sample facilities was 83.3%. Two hospitals recorded a score of 100% and the other recorded a score of 67.7%. The average diagnostic capacity readiness score of the general hospital in the study sample was 96.9%. The average score for essential medicine readiness was 90%. The average general service readiness score of the study facilities was 88%. General Hospital Bida had a score of 72%, General Hospital Kontagora—97%, Minna—97%, and Suleja—87%.

Spearman’s correlation was performed between the overall satisfaction level with the healthcare care that was received, and the five domains of health facility readiness and the general facility readiness (see Table 2). No statistically significant association was observed between these factors. In other words, the level of preparation in any of the domains, as well as the general facility readiness, did not show any relationship with how satisfied the patients were with their healthcare experiences.

### 3.3. The Concept of the Healthcare Process with Patient Satisfaction

There is no consensus in the literature on how to define the concept of patient satisfaction in healthcare. For satisfaction, more than half of the respondents (disagree 30%, strongly disagree 25.3%) disagree that doctors are good at explaining the reason for medical tests (see Figure 2). A total of 543/734 (74%) of the respondents disagree that the doctor’s office contains everything that is needed to provide complete medical care. About 38% of participants strongly agree and about 42% disagree, respectively, that their medical bills are often beyond their reach. An overwhelming 639 (87%) of the respondents either strongly agree or agree that when they need emergency care, the waiting times are usually too long. Regarding the time that the multimorbid patients spent with the doctor, less than half of the respondents 298 (40.6%) were satisfied, while the remaining portion were not satisfied with the doctor–patient time, and nine (1.2%) of the respondents were not certain or remained indifferent to the doctor–patient time (see Figure 2).

The findings from Table 3 show that in the unadjusted multiple regression model for the overall satisfaction with the healthcare care that is received and the patient health experiences variables, all the variables of patient’s healthcare experiences are predictors of overall patient satisfaction, except the variable that sometimes doctors make the patients wonder if their diagnosis is correct (see Table 4). In the adjusted model, eight of the variables remain significant. The strength of the remaining prediction variables is reduced to varying degrees. Overall, in the unadjusted model, the coefficient is higher in the variable that those who provide patients’ medical care sometimes hurry too much when they treat them, but in the adjusted model, it is higher in the variable that patients can receive medical care whenever they need it. These variables reflect the patients’ quality of care and access to medical care.

A Principal Components Analysis (PCA) was performed to reduce the dimension or group of the principal factors of the process indicator (patient experience) dataset among the multimorbid patients in total. Four eigenvalues were observed and collectively accounted for 53.450% of the variation in patients’ experience satisfaction (see Table 4). Seven questions that loaded onto the first component related to the accessing quality of care and six questions that loaded onto the second component related to patient–physician relationship and time, whereas two questions that loaded onto the third component related to the financial burden of medical care. The remaining three questions that loaded onto the fourth component related to the confidence and trust in medical care. Thus, in brief, the PCA shows that (1) accessing quality care; (2) patient–physician relationship and time; (3) the financial burden of medical care; (4) confidence and trust in medical care are the principal factors in the study that influence multimorbid patient experience satisfaction.

## 4. Discussion

There are different frameworks for measuring the quality of care, the most common being the World health organization (WHO) recommended quality of care framework, the Bamako initiative, and the Donabedian model. Although each of these frameworks of measuring quality of care is not devoid of disadvantages, this study adopted the Donabedian model of care. This is chiefly because the Donabedian model is not only a conceptual model that provides a framework for examining health services and evaluating the quality of health care [16], but also because it gathers information on the quality of care through structure, process, and outcome [17] (see Figure 1). Having established this, it is possible to consider that the framework can be used to modify structures and processes within a healthcare delivery unit, such as a small group practice or ambulatory care center, in order to improve patient flow or information exchange [18]. The healthcare provider–user interface that is reported in this study substantiates the flexible application in diverse healthcare settings that can be used to modify structures, processes, and outcomes within a healthcare delivery unit, as postulated by Donabedian [17,19,20].

While the general service readiness index for this study was 83.3%, it was not uniform across the board, with health facilities having moderate to high scores (GH Bida—72%; GH Kontagora—97%; GH Minna—97%; and GH Suleja—87%). It is only in the domain of basic equipment that all the tracer items are complete in the four health facilities. This study found a shortage of diagnostic capacity which partially coincides with findings elsewhere [21]. Further, such a scarcity of diagnostic tests limits the ability of healthcare providers to offer quality care. No statistically significant association was observed between the facility readiness domains, as well as the general service readiness index, and the overall patient satisfaction. This is surprising; however, it can be explained, as around two-thirds of the participants in the study setting lack any form of education, their understanding of standard requirements for health facilities may be limited, and their current experiences may be outside of their expectations. Another consideration is that, despite the low literacy level among participants, the study clearly demonstrated poor accommodation of patients’ needs, evidenced by the level of dissatisfaction that was reported with variables of patients’ experience.

The present study found that satisfaction with the variables of patients’ processes of care was low, which is similar to [22], which observed that overall achievement for the process of care-related quality indicators was also poor. Regarding correlation with overall satisfaction, the present study reported findings that were similar to [22,23], which reported experiences suggesting that the process of certain aspects of care explained most of the variance in the overall assessment, measured in terms of a global rating. Further, process proved to be the most important predictor, rather than structure and outcome.

However, researchers have argued that the Donabedian model has faults, because the sequential progression from structure to process to outcome is, as described by some, too linear a framework [24], and consequently has a limited utility for recognizing how the three domains influence and interact with each other [25]. Another issue of importance is the failure to incorporate antecedent characteristics (e.g., patient characteristics, environmental factors) which are important precursors to evaluating quality of care. Overall, one major advantage of the Donabedian model is that it is a quality-of-care framework model that is developed to be flexible enough for application in diverse healthcare settings and among various levels within a delivery system.

However, an evaluation of manpower was not part of the structure assessment for this study. It is appropriate to mention that the health system in Nigeria is currently suffering not only a limited institutional capacity, but also the worst brain drain in the history of the country. This is happening in the face of daily societal issues such as a lack of security, banditry, kidnapping, political instability, corruption, an unstable economy, and worsening health indexes. In some studies, the perceived lack of essential drugs that is observed by respondents is crucial, because the availability of essential drugs is an important factor influencing patients’ level of satisfaction observed in several studies in other settings [26,27,28]. This is partly contrary to our finding that reported essential drugs were available at 90%.

The process evaluation was by patient experience with healthcare services. Using a principal component analysis, the process items were reduced into four main components (i) accessing the quality of care; (ii) patient–physician relationship and time; (iii) the financial burden of the medical care; (iv) confidence and trust in the medical care. This is partially consistent with the findings of a study in Mwananyamala Hospital in Dar es Salaam, Tanzania, that used the Principal Component Analysis (PCA) to identify six items (three empathy items versus three tangible items) that explained 53 percent of the patient’s satisfaction scores on quality of care [29]. However, the identified components in their studies were empathy items, which explained why most of the dissatisfaction with the quality of care was related to a failure to show compassion, a lack of politeness, and inadequate listening by OPD staff. However, it is important to note that factors that influence patient satisfaction and quality of care are multifactorial, and thus caution should be taken when making conclusions regarding quality of care [30,31,32].

According to [33], easy-to-navigate pathways to care and continuity are critical to how patients perceive the quality of care and choose whether to continue treatment or not. The authors further state that long-term compliance is only likely if the patients that are involved consider their care to be of a good quality. The greatest factor that influences the patients’ overall satisfaction with the quality of services is the variable of those who provide their medical care sometimes being in a hurry (quality issues) in the unadjusted model, but being able to receive medical care whenever they need it (access issue) in the adjusted model. These variables are reflecting the patients’ quality of care and access to medical care. Moreover, paying attention to these main factors is essential in designing effective quality health care for multimorbid patients. Lastly, what patients think of their experience with the healthcare system must matter to healthcare planners, managers, and policymakers, because patients’ experience, as much as the technical quality of care, determines how people use the system and how they benefit from it [34].

## 5. Strength and Limitations

Although the result of this study has the potential to illuminate some of the weaknesses of the current multimorbidity care among the elderly, the sample selection is limited to four hospitals in Niger State; thus, the findings cannot be generalized to Nigeria. Nevertheless, the study can be replicated elsewhere in the country to increase its impact. The strength of this study is grounded in the context that it used the Donabedian model, which has been tested in many studies on patient satisfaction, revealing significant results. The model is a direct target for quality improvement. However, only selected tracer items were used and the model only focused on the patient’s perception of their satisfaction with their experiences as they navigate the care pathway.

## 6. Conclusions

It could be concluded that the process component superseded the structure as the determinant of the quality of healthcare among multimorbid patients in Niger State. Further, the process’s emphasis should be on improving access to quality of care, improving patient–physician relationships and time, reducing the financial burden of medical care, and building confidence and trust in medical care. Therefore, it should be incorporated into designing the healthcare model for multimorbid patients in Nigeria. This report is particularly important to better inform policymakers and related stakeholders, in order to ensure equitable access and improve the health outcomes of multimorbid patients and the overall population’s health.

## Figures and Tables

**Figure 1 healthcare-10-01268-f001:**
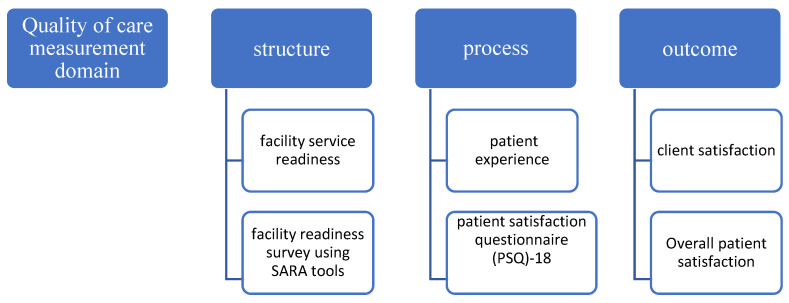
Illustrating the Donabedian model of care.

**Figure 2 healthcare-10-01268-f002:**
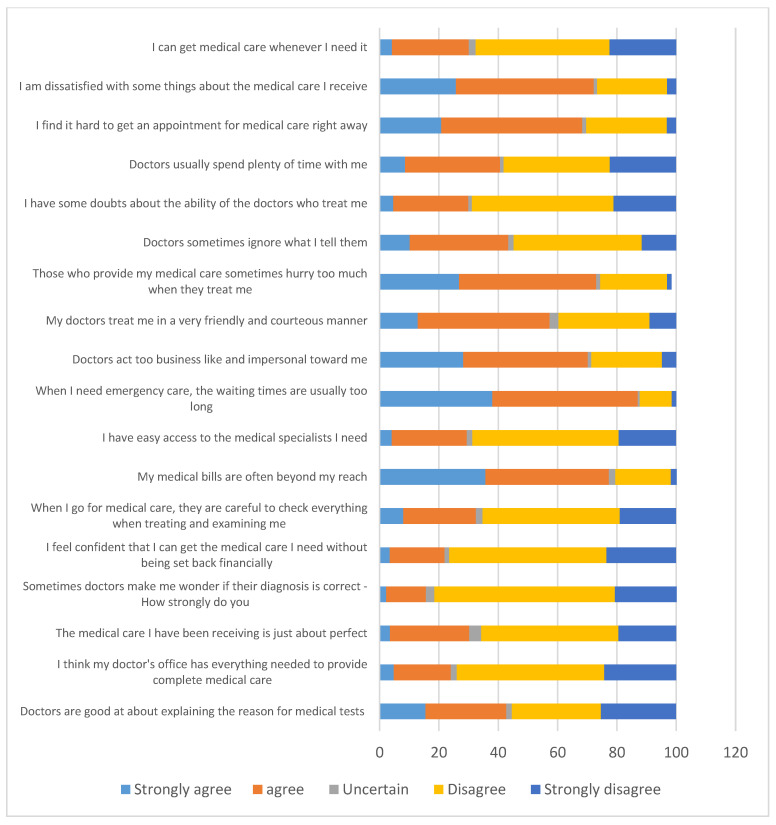
Patient satisfaction with quality of services among older people in Nigeria with multimorbidities.

**Table 1 healthcare-10-01268-t001:** Socio-demographic characteristics of the respondent. (*n* = 734). * Mean age.

Variables	*n*	%
Gender		
Male	300	40.9
Female	434	59.1
Total	734	100
Age * 67.37 (66.37 for male and 68.06 for female)		
60–64	262	35.7
65–69	267	36.4
70–74	123	16.8
75–79	29	4.0
80 and greater	53	7.2
Total	734	100.0
Marital status		
Never married	11	1.5
Currently married	483	65.8
Divorced	21	2.9
Separated	19	2.6
Widow/er	200	27.2
Total	734	100.0
Family structure		
Nuclear family	140	19.1
Three-generation family	150	20.5
Extended family	442	60.4
Total	732	100.0
Education level		
Illiterate	462	62.9
Can read and write	35	4.8
Primary school level	74	10.1
Secondary school	64	8.7
Tertiary school	83	11.3
Post-graduate	16	2.2
Total	734	100.0
Occupation		
Government staff	36	4.9
Own business	280	38.1
Involve in the family business	36	4.9
Company staff/worker	30	4.1
Dependent	214	29.2
Retired	128	17.4
Others (specify)	10	1.4
Total	734	100
Ethnicity		
Gwarri	193	26.3
Hausa	174	23.7
Nupe	204	27.8
Others	163	22.2
Total	734	100
Level of income		
NGN 0–15k	477	65.0
NGN 16–30k	124	16.9
NGN 31–45k	30	4.1
NGN 46–60k	27	3.7
Greater than NGN 60	76	10.4
Total	734	100

**Table 2 healthcare-10-01268-t002:** Mean availability of items by domain score and satisfaction level with the healthcare received and facility readiness.

	Number of Available Items (Mean Score)					The Overall Satisfaction Level with the Healthcare Received	
Domains	GH Bida	GH Kontagora	GH Minna	GH Suleja	Total	Rho	*p*-Value
Basic amenities	2 (28.6%)	6 (85.7%)	6 (85.7%)	6 (85.7%)	71.4%	0.272	0.728
Basic equipment	6 (100%)	6 (100%)	6 (100%)	6 (100%)	100%	0.123	0.635
Standard precautions for infection prevention	6 (66.7%)	9 (100%)	9 (100%)	6 (66.7%)	83.3%	−0.236	0.764
Diagnostic capacity	8 (100%)	8 (100%)	8 (100%)	7 (87.5%)	96.9%	−0.544	0.456
Essential medicines	13 (65%)	20 (100%)	20 (100%)	19 (95%)	90%	0.500	0.500
General service readiness index = (mean score of the five domains) (a + b + c + d + e)/5	72%	97%	97%	87%	88.3%	0.211	0.789

**Table 3 healthcare-10-01268-t003:** Multiple regression model for overall satisfaction level with the medical care received and patient’s health care experiences.

	Unadjusted Coefficient (b)	*p*-Value	Adjusted Coefficient (b)	*p*-Value
Doctors are good at explaining the reason for medical tests	−0.407 **	0.001	−0.026	0.524
I think my doctor’s office has everything needed to provide complete medical care	−0.322 **	0.001	−0.075 *	0.027
The medical care I have been receiving is just about perfect	−0.328 **	0.001	0.040	0.275
Sometimes doctors make me wonder if their diagnosis is correct	−0.060	0.102	x	
I feel confident that I can get the medical care I need without being set back financially	−0.210 **	0.001	0.027	0.421
When I go for medical care, they are careful to check everything when treating and examining me	−0.452 **	0.001	−0.240 *	0.002
My medical bills are often beyond my reach	0.135 **	0.001	0.002	0.949
I have easy access to the medical specialists I need	−0.255 **	0.001	0.038	0.232
When I need emergency care, the waiting times are usually too long	0.324 **	0.001	−0.024	0.455
Doctors act too business-like and impersonal toward me	0.502 **	0.001	0.252 **	0.001
My doctors treat me in a very friendly and courteous manner	−0.322 **	0.001	−0.057	0.080
Those who provide my medical care sometimes hurry too much when they treat me	0.519 **	0.001	0.142 **	0.001
Doctors sometimes ignore what I tell them	0.274 **	0.001	0.089 *	0.004
I have some doubts about the ability of the doctors who treat me	0.133 **	0.001	0.090 *	0.002
Doctors usually spend plenty of time with me	−0.350 **	0.001	−0.036	0.279
I find it hard to get an appointment for medical care right away	0.381 **	0.001	0.062 *	0.050
I can get medical care whenever I need it	−0.482 **	0.001	−0.240 **	0.001

Correlation is significant at 0.05 *. Correlation is significant at the 0.01 ** level (2-tailed).

**Table 4 healthcare-10-01268-t004:** Principal component analysis (PCA) summarizing patient experiences.

Component	Initial Eigenvalues			Extraction Sums of Squared Loadings			Factor Loading and Commonalities for Independent Variables			
	Total	% of Variance	Cumulative %	Total	% of Variance	Cumulative %	1	2	3	4
I think my doctor’s office has everything needed to provide complete medical care	5.361	29.786	29.786	5.361	29.786	29.786	0.757	0.019	−0.043	0.017
Doctors are good at explaining the reason for medical tests	1.633	9.074	38.860	1.633	9.074	38.860	0.753	−0.263	0.010	−0.098
The medical care I have been receiving is just about perfect	1.505	8.361	47.221	1.505	8.361	47.221	0.730	−0.130	0.200	0.033
When I go for medical care, they are careful to check everything when treating and examining me	1.121	6.229	53.450	1.121	6.229	53.450	0.633	−0.414	0.142	−0.031
My doctors treat me in a very friendly and courteous manner	0.971	5.395	58.845				0.578	−0.209	0.071	−0.168
I have easy access to the medical specialists I need	0.909	5.052	63.897				0.447	−0.101	0.380	0.005
I can get medical care whenever I need it	0.842	4.677	68.574				0.435	−0.373	0.280	0.045
Doctors act too business-like and impersonal toward me	0.729	4.051	72.625				−0.185	0.797	−0.092	0.046
Those who provide my medical care sometimes hurry too much when they treat me	0.681	3.784	76.409				−0.226	0.764	−0.124	0.204
When I need emergency care, the waiting times are usually too long	0.669	3.718	80.127				0.085	0.647	−0.194	−0.045
I am dissatisfied with some things about the medical care I receive	0.587	3.263	83.390				−0.353	0.610	−0.039	0.243
Doctors usually spend plenty of time with me	0.571	3.173	86.563				0.325	−0.550	−0.052	0.230
I find it hard to get an appointment for medical care right away	0.515	2.860	89.423				−0.147	0.526	−0.139	0.078
My medical bills are often beyond my reach	0.451	2.506	91.928				0.082	0.174	−0.825	0.062
I feel confident that I can get the medical care I need without being set back financially	0.408	2.267	94.196				0.282	−0.172	0.683	0.028
I have some doubts about the ability of the doctors who treat me	0.393	2.184	96.379				−0.141	−0.053	−0.012	0.796
Doctors sometimes ignore what I tell them	0.346	1.920	98.299				−0.095	0.274	−0.125	0.595
Sometimes doctors make me wonder if their diagnosis is correct	0.306	1.701	100.000				0.234	−0.018	0.372	0.508

Kaiser-Meyer-Olkin Measure of Sampling Adequacy—0.872. Bartlett’s Test of Sphericity—3873.187. *p*-value—0.000. Extraction Method: Principal Component Analysis.

## Data Availability

Not applicable.

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
