# Peer review of "Effective Hospital Care Delivery Model for Older People in Nigeria with Multimorbidity: Recommendations for Practice"

_healthcare, 2022, doi:10.3390/healthcare10071268_

Round 1

Reviewer 1 Report

The paper fills a gap on patient reported experience with health care in Nigeria. It surprisingly finds no correlation between structure of health care and patient reported satisfaction. However, the result should be interpreted in the context of the Nigerian health care system.

1.     It would be useful to describe the system briefly in the Introduction. Are patients insured? Publicly/privately? Are providers public? How accessible are hospitals? Etc. That would help readers not familiar with Nigeria understand the research better.

2.     Please add to Abstract when the study was conducted, in what setting (hospital?), rural/urban setting, and other details.

3.     Please move parts discussing your choice for research methodology to Discussion from Introduction (page 2, lines 55-74).

4.     Please revise Figure 1 – it seems that the lowest line of blocks shows your design, rather than the model of care itself. Please relabel figure as necessary to reflect what it shows.

5.     Methods would benefit from major revision. Please add details of how the interviews were conducted, if the participants were compensated. Please provide the questionnaires (in English) in Appendix. It is not clear how consent was obtained 24 hours in advance, if patients were approached immediately after the consultation. By phone in advance? How the team got the contact details of the patients in advance in that case? It would be great to clarify this.

6.     It is not clear what “facility survey with standard tools” refers to. What are “standard tools”?

7.     Please describe how multimorbidity was measured and assessed. Self-reported? Patient charts? Any other method?

8.     Please describe the 4 health facilities where research took place. It seems that they are named in one of the tables. If you identify them in the article, please describe (setting: rural/urban, size, etc.)

9.     Results need to be shortened significantly. There is no need to state in text what is shown in tables. It is difficult to follow the very long text of Results. E.g., page 4, first paragraph could be practically removed. You can also significantly shorten the details about testing for normality (e..g, just mention that assumptions were met for linear regression). If you choose to detail, please revise to make sure necessary details are given (e.g., line 168 – scatter plot of which two variables). Please also make sure to name outcome variable and predictors for all regressions.

10.  Results mention Table 7.4 I could not find it.

11.  Please review references. Some refer just to author and date – please add them to the reference list.

12.  Please add to Intro and Discussion comments comparing your study to any similar studies. This is the first one in Nigeria, but maybe in other similar countries anything similar was already done. Currently, there is virtually no literature context provided (besides describing the framework).

13.  Move first sentence of Results to Methods.

14.  Please explain how much is “naira” and add it to Table (currently, only amount, not currency of income provided).

15.  Please revise Tables to format them in a standard way, with subheadings, right or left alignment. It is difficult to follow the information now.

16.  Please revise all text to remove repetitive statements. E.g., Donabedian model is described similarly maybe 3 or more times.

17.  Table 3: no need for last column. Consider reformatting as a graph. It is difficult to read so many numeric results.

18.  Please shorten results on page 9. It is very hard to follow, please focus on the main findings and direct readers to Tables for more details.

19.  Table 4 spans 3 pages. That is probably too long. It also contains multiple testing without adjustment. Please consider moving most of it to an Appendix.

20.  Please strongly consider putting results from line 299 to Appendix. They are not discussed and seem rather a methodological comment rather than adding substantive knowledge.  

21.  Pleas revise Strengths and limitations to be more specific.

22.  Please revise the text. It seems that in some places wrong words are used (e.g., patience instead of patients?). Some sentences seem incomplete.

23.  Please revise to avoid unfounded or exaggerated statements (“mystery why there is no research”, “there is no relationship” – when merely no correlation is shown in your particular study; “anonymous” instead of “unknown”).

Author Response

First reviewer’s comments

Manuscript I.D healthcare-1776966

Response to Reviewers

Dear Reviewer,

Thank you for giving us the opportunity to submit a revised draft of the manuscript “To recommend an effective hospital care delivery model for older people in Nigeria with multimorbidity”. We appreciate the time and effort that you and the reviewers dedicated to providing feedback on our manuscript and are grateful for the insightful comments on and valuable improvements to our paper. We have incorporated most of the suggestions made by the reviewers. Those changes are corrected within the manuscript. Please see below, in red, for a point-by-point response to the reviewers’ comments and concerns. All page numbers refer to the revised manuscript file with tracked changes.

  1. Reviewer comment to the author

The paper fills a gap on patient-reported with health care in Nigeria. It surprisingly finds no correlation between structure of health care and patient reported satisfaction. However, the result should be interpreted in the context of the Nigerian health care system.

Authors response

We thank the reviewer for pointing this out. However, we have revised it in the context of Nigeria.

We’ve changed “By focusing more on this in more detail, this is no surprise because the majority of the participants lack any form of education, and their understanding of the domains of the health facility, as well as the general facility readiness may be limited and their current experience maybe be beyond their expectations.  to “This is surprising, however, by focusing more on this in more detail, this can be explained because about two-thirds of the participants in the study setting lack any form of education, and their understanding of the domains of the health facility, as well as the general facility readiness may be limited and their current experiences maybe are beyond their expectations.”. (This was discussed on page 11, lines 319- 323).

  1. Reviewer comment to the author

It would be useful to describe the system briefly in the Introduction. Are patients insured? Publicly/privately? Are providers public? How accessible are hospitals? Etc. That would help readers not familiar with Nigeria understand the research better.

Authors response

Thank you for pointing this out. The reviewer is correct. We agree and have updated to include the healthcare system of the study area accordingly in the introduction session as follows

Nigeria operates a pluralistic healthcare delivery system (orthodox and traditional healthcare delivery systems). Orthodox health care services are provided by the private and public sectors. However, the provision of health care in Nigeria remains the function of the three tiers of government: the federal, state and local government (Page 2, lines 58-61)

  1. Reviewer comment to the author

Please add to the Abstract when the study was conducted, in what setting (hospital?), rural/urban setting, and other details.

Authors response

As suggested by the reviewer, we have made the changes to include study setting details in the abstract and the new sentence reads as follows

A cross-sectional study using face-to-face data was conducted between October 2021 to February 2022. Reported data were collated, checked, coded, and entered into JISC online survey software and were exported to Statistical Package for Social Science (SPSS) version 27 for analysis. Data were collected from the outpatient department of 4 high-volume public secondary hospitals in Ni-ger state (the largest hospital in the 3 senatorial zones and that of the state capital). Discussed in page 1, lines 17-21

  1. Reviewer comment 4 to the author

Please move parts discussing your choice for research methodology to Discussion from Introduction (page 2, lines 55-74).

Authors response

Thank you, reviewer, for this suggestion. We have moved the part discussing research methodology from the introduction to the discussion. The part is now discussed on page 10 lines 295- 308 and on page 11 lines 334-342

  1. Reviewer comment to the author

Please revise Figure 1 – it seems that the lowest line of blocks shows your design, rather than the model of care itself. Please relabel the figure as necessary to reflect what it shows.

Figure 1 has been relabeled to reflect the idea considered. The figure is now been labeled as

Figure 1 Illustrating the Donabedian model of care (page 3)

  1. Reviewer comment to the author

Methods would benefit from major revision. Please add details of how the interviews were conducted, if the participants were compensated. Please provide the questionnaires (in English) in Appendix. It is not clear how consent was obtained 24 hours in advance if patients were approached immediately after the consultation. By phone in advance? How the team got the contact details of the patients in advance in that case? It would be great to clarify this.

The methods were revised to include how the interview was conducted, and an explanation was provided on 24 hours gap between being provided with information about the study and being involved in interviews. It was also stated that participants were not compensated. The questionnaire in English is attached to the appendix. As discussed on page 3 lines 99-137

  1. Reviewer comment to the author

It is not clear what “facility survey with standard tools” refers to. What are “standard tools”?

Authors response

Thank you for the observation. The authors meant to use the word comprehensive Service Availability and Readiness Assessment (SARA) tool; however, it has been changed accordingly in the revised submission. As discussed in the abstract and measurements of variables on Page 1 line24 and page 4 line 144 respectively.

  1. Reviewer comment to the author

            Please describe how multimorbidity was measured and assessed. Self-reported?                 Any other method?

Authors response

We thank the reviewer for pointing this out. We have revised how multimorbidity was measured and assessed described as

“Although the choice of the measure of multimorbidity is mainly on the suitability of the measure for data obtainable and the predilection of the researcher, the earlier cited systemic review shows that the most common approach to measuring multimorbidity is the use of simple disease counts. And they concluded that simple measures, such as counts of chronic diseases, are almost as effective at predicting health care utilization and quality of life as more sophisticated measurements. On account of the aforementioned reasons, this study uses a simple disease count” page 2 lines 72-79 lines

  1. Reviewer comment to the author

Please describe the 4 health facilities where research took place. It seems that they are named in one of the tables. If you identify them in the article, please describe (setting: rural/urban, size, etc.)

Authors response

The 4 health facilities where the research took place were described as

The participants were uninsured and recruited from the outpatient department of 4 high-volume public secondary hospitals in Niger state namely general hospital Minna, general hospital Bida, general hospital Suleja and general hospital Kontagora. While the secondary hospital Minna is the largest in the state the remaining three represent the largest hospitals each from the 3 largest towns in senatorial A, B, and C zones respectively. These 4 hospitals combined see about 85 to 90% of patients load in the state. Multimorbidity challenges existing healthcare organizations, research, family relationships, and social security, chiefly because the healthcare organization remains a single-disease condition focused. As discussed on page 3 lines 107 -115

  • Reviewer comments to the author

Results need to be shortened significantly. There is no need to state in text what is shown in tables. It is difficult to follow the very long text of Results. E.g., page 4, first paragraph could be practically removed. You can also significantly shorten the details about testing for normality (e..g, just mention that assumptions were met for linear regression). If you choose to detail, please revise to make sure necessary details are given (e.g., line 168 – scatter plot of which two variables). Please also make sure to name outcome variable and predictors for all regressions.

Authors response

We thank the reviewer for pointing this out. We have revised as follows

The statistical analysis has been shortened significantly to improve comprehension. As suggested some confusing paragraphs have been removed. The detailed assumptions were adequately summarized. The results on page 4 lines 160-176, page 5 lines 180-189 of the initial manuscript were removed.

  1. Reviewer comments to the author

Results mention Table 7.4 I could not find it.

Authors response

Thank you for pointing this out. It’s a typing error and has been corrected accordingly 

  1. Reviewer comments to the author

Please review references. Some refer just to author and date – please add them to the reference list.

Authors response

Thank you for a very important observation. The references were updated checked for completion, corrections were made and consistency was maintained accordingly.  

  1. Reviewer comments to the author

Please add to Intro and Discussion comments comparing your study to any similar studies. This is the first one in Nigeria, but maybe in other similar countries anything similar was already done. Currently, there is virtually no literature context provided (besides describing the framework).

Authors response

Comparison to other studies was added to both the introduction and discussion sessions. As discussed on page 2, lines 68-70, 80-92, and on page 11, line 314, 328- 333, 346 -351 respectively.

  1. Reviewer comments to the author

Move first sentence of Results to Methods.

Authors response

The first sentence of the results has been moved to the methods session as suggested. It is now in page 4 lines 160-161

  1. Reviewer comments to the author

Please explain how much is “naira” and add it to Table (currently, only amount, not currency of income provided).

Authors response

We thank the reviewer for pointing this out. We have revised it and has been added to the table

accordingly. As discussed on page 5, lines 201-203

  1. Reviewer comments to the author

Please revise Tables to format them in a standard way, with subheadings, right or left alignment. It is difficult to follow the information now.

Authors response

All Tables were reformatted with subheading to the left.

  1. Reviewer comments to the author

Please revise all text to remove repetitive statements. E.g., Donabedian model is described similarly maybe 3 or more times.

Authors response

Thank you for this suggestion. Repetitive statements removed as suggested

  1. Reviewer comments to the author

Table 3: no need for last column. Consider reformatting as a graph. It is difficult to read so many numeric results.

Authors response

The last column was removed and reformatted as a bar chart graph and labeled as figure 2 on Page 8

  1. Reviewer comments to the author

Please shorten results on page 9. It is very hard to follow, please focus on the main findings and direct readers to Tables for more details.

Authors response

As suggested the result on page 9 has been significantly shortened, maintaining the value of the results. Only the main findings are included and references are made to the table appropriately.

  1. Reviewer comments to the author

Table 4 spans 3 pages. That is probably too long. It also contains multiple testing without adjustment. Please consider moving most of it to an Appendix.

Authors response

Table 4 has been reviewed. The table was reformatted to maintain only elements needed for the interpretation in the context of the study. The revised table is labeled Table 3 on page 9.

  1. Reviewer comments to the author

Please strongly consider putting results from line 299 to Appendix. They are not discussed and seem rather a methodological comment rather than adding substantive knowledge.  

Authors response

The results on this line have been reviewed and only the most important aspects are retained.

  1. Reviewer comments to the author

Please revise the Strengths and limitations to be more specific.

Authors response

The strength and limitations have been revised to be more specific to the study.  As discussed on page 12 lines 381-389

  1. Reviewer comments to the author

Please revise the text. It seems that in some places wrong words are used (e.g., patience instead of patients?). Some sentences seem incomplete.

      Authors response

The text has been reviewed thoroughly involving an expert.

  1. Reviewer comments to the author

Please revise to avoid unfounded or exaggerated statements (“mystery why there is no research”, “there is no relationship” – when merely no correlation is shown in your particular study; “anonymous” instead of “unknown”).

     Author response

The text has been reviewed considerately. The word mystery has been substituted with anonymous. Page 1 line 14 and page 2 line 91

Reviewer 2 Report

Dear Authors and thanks for the opportunity to review this paper. Manuscript intention seems to be interesting, however, the paper is presented in a confusing way and a broad english editing is necessary.

Abstract should be unstractured (please see author instruction).

The main text is not clear presented and confused. The aim is not well explained it needs to be concise. Methods should be well describe and logically expressed. Authors used Principal Component Analysis (it is an old methodology, please explain why you decided to use it). Authors spend a lot of words to explain the methodology used, such as the definition of linear regression. Results are also not concise and full of elements (tables and figures), which could be organized differently. Conclusion is approximately discusses.

I have the impression that authors wanted to present many elements, I think it is better to present fewer elements but better organized. 

Author Response

Second reviewer’s comment

Manuscript I.D healthcare-1776966

Response to Reviewers

Dear Reviewer,

Thank you for giving us the opportunity to submit a revised draft of the manuscript “To recommend an effective hospital care delivery model for older people in Nigeria with multimorbidity”. We appreciate the time and effort that you and the reviewers dedicated to providing feedback on our manuscript and are grateful for the insightful comments on and valuable improvements to our paper. We have incorporated most of the suggestions made by the reviewers. Those changes are corrected within the manuscript. Please see below, in red, for a point-by-point response to the reviewers’ comments and concerns. All page numbers refer to the

.

  1. Reviewer comment to the author

The abstract should be unstructured (please see author instruction).

Authors response

Thank you for drawing our attention. The abstract has now been revised to an unstructured format. See page 1

  1. Reviewer comment to the author

The main text is not clearly presented and confused.

Authors response

The main text has been largely reviewed to improve comprehension.

  1. Reviewer comment to the author

The aim is not well explained it needs to be concise.

Authors response

Thank you for that observation. In the revised manuscript the aim of the study is clearly stated in a concise fashion. See page 2 lines 93- 94.

  1. Reviewer comment to the author

Methods should be well describe and logically expressed.

Authors responses

The methods were described in more detail and rearranged with a more logical expression in the revised manuscript. As discussed on page 3, lines 99 -130

  1. Reviewer comment to the author

Authors used Principal Component Analysis (it is an old methodology, please explain why you decided to use it).

Authors response

Principal component analysis was used to determine the dimension of the patient experience as they navigate the care pathway.

  1. Reviewer comment to the author

Authors spend a lot of words to explain the methodology used, such as the definition of linear regression.

Authors response

Thank you for that observation. Explanation text have been removed significant making the article more direct and concise

  1. Reviewer comment to the author

Results are also not concise and full of elements (tables and figures), which could be organized differently.

Authors response

The result on page 9 has been significantly shorten, maintaining references to the results. Only the main findings are included and reference are made to the table appropriately.

  1. Reviewer comment to the author

Conclusion is approximately discussed.

Authors response

Thank you.

  1. Reviewer comment to the author

I have the impression that authors wanted to present many elements, I think it is better to present fewer elements but better organized. 

Authors response

Thank you for your observation and suggestion. The manuscript has been revised sufficiently to only retain key elements most relevant to the research.

Round 2

Reviewer 2 Report

Dear authors, thanks for the changes made. The article has been very well implemented and edited. There still remain a few things that in my opinion need to be improved and specified, such as:

- Figure 2 lacks specific item references;

- The PCA is an old methodology, the rationale on why it was conducted compared to other types of analysis needs to be clarified;

- The objective of the research is still not very clear, I ask for more structured detail. 

- Pay attention to the various typos in the text (e.g., the word title should not appear)

Author Response

Manuscript I.D healthcare-1776966

Response to Reviewers

Dear Reviewer,

Thank you for giving us the opportunity to submit a revised draft of the manuscript “To recommend an effective hospital care delivery model for older people in Nigeria with multimorbidity”. We appreciate the time you dedicated to providing feedback on our manuscript and are grateful for the insightful comments and valuable improvements to our paper. We have incorporated most of the suggestions made by the reviewers. Those changes are corrected within the manuscript. Please see below, in red, for a point-by-point response to the reviewers’ comments and concerns. All page numbers refer to the

  1. Reviewer’s comments to Authors

Figure 2 lacks specific item references;

Author’s response

Thank you for this observation. It has been corrected accordingly. By adding additional information to make it more specific.

  1. Reviewer’s comments to Authors

The PCA is an old methodology, the rationale on why it was conducted compared to other types of analysis needs to be clarified;

Author’s response

Thank you for this comment. The rationale for its selection has been added accordingly. As discussed on page 5 lines 189-194

  1. Reviewer’s comments to Authors

The objective of the research is still not very clear, I ask for more structured detail. 

Author’s response

Thank you for your observation. The research objective has been rewritten and we hope it meets your expectations. As discussed on page 2 lines 92-95

  1. Reviewer’s comments to Authors

        Pay attention to the various typos in the text (e.g., the word title should not appear)

          Author’s response.

      Thank you for this observation. More attention was given to typos in the text. The word title has been removed accordingly.
